# Prevalence and incidence of possible vascular dementia among Mexican older adults: Analysis of the Mexican Health and Aging Study

**Sara G. Yeverino-Castro**[1☯], **Silvia Mejía-Arango**[2☯], **Alberto J. Mimenza-Alvarado**[1,3‡], **Carlos Cantú-Brito**[4‡], **José A. Avila-Funes**[3,5‡], **Sara G. Aguilar-Navarro**[1,3]*

1 Geriatric Medicine & Neurology Fellowship, Instituto Nacional de Ciencias Médicas y Nutrición Salvador Zubiran, Mexico City, Mexico, 2 Department of Population Studies, El Colegio de la Frontera Norte, Tijuana, Baja California, México, 3 Department of Geriatric Medicine, Instituto Nacional de Ciencias Médicas y Nutrición Salvador Zubirán, Mexico City, Mexico, 4 Department of Neurology and Psychiatry, Instituto Nacional de Ciencias Médicas y Nutrición Salvador Zubirán, Mexico City, Mexico, 5 Inserm, Bordeaux Population Health Research Center, UMR 1219, Univ. Bordeaux, Bordeaux, France

☯ These authors contributed equally to this work.
‡ AJM-A, CC-B and JAA-F also contributed equally to this work.
* sgan30@hotmail.com

**Data Availability Statement:** Data are available from the The Mexican Health and Aging Study

## Abstract

### Introduction

Vascular dementia is the second most common cause of dementia. Physical disability and cognitive impairment due to stroke are conditions that considerably affect quality of life. We estimated the prevalence and incidence of possible vascular dementia (PVD) in older adults using data from the Mexican Health and Aging Study (MHAS 2012 and 2015 waves).

### Methods

The MHAS is a representative longitudinal cohort study of Mexican adults aged ≥50 years. Data from 14, 893 participants from the 2012 cohort and 14,154 from the 2015 cohort were analyzed to estimate the prevalence and incidence of PVD. Self-respondents with history of stroke were classified as PVD if scores in two or more cognitive domains in the Cross-Cultural Cognitive Examination were ≥ 1.5 standard deviations below the mean on reference norms and if limitations in ≥ 1 instrumental activities of daily living were present. For proxy respondents with history of stroke, we used a score ≥3.4 on the Informant Questionnaire on Cognitive Decline in the Elderly. Crude and standardized rates of prevalent and incident PVD were estimated.

### Results

Prevalence of PVD was 0.6% (95% CI, 0.5–0.8) (0.5 with age and sex- standardization). Rates increased with age reaching 2.0% among those aged 80 and older and decreased with educational attainment. After 3.0 years of follow-up, 87 new cases of PVD represented an overall incident rate of 2.2 (95% CI, 1.7–2.6) per 1,000 person-years (2.0 with age and

(MHAS) from www.mhasweb.org (Database: https://doi.org/10.1093/ije/dyu263) for researchers who meet the criteria for access to confidential data.

**Funding:** This work was supported by the Consejo Nacional de Ciencia y Tecnología (CONACYT) (Mexican National Counsel of Science and Technology ((FOSISS 2017-1 290406 2017. GER-2416-18-20-1).

**Competing interests:** The authors have declared that no competing interests exist.

sex- standardization). Incidence also increased with advancing age reaching an overall rate of 9.4 (95% CI, 6.3–13.6) per 1,000 person-years for participants aged >80 years. Hypertension and depressive symptoms were strong predictors of incident PVD.

## Conclusion

These data provide new estimates of PVD prevalence and incidence in the Mexican population. We found that PVD incidence increased with age. Males aged 80 years or older showed a greater incidence rate when compared to females, which is comparable to previous estimates from other studies.

## Introduction

Vascular Cognitive Impairment (VCI) is the second most common cause of dementia, with 15% of dementia cases [1]. Post-stroke physical disability and cognitive impairment are conditions that considerably affect quality of life. [2]. When compared to Alzheimer's disease (AD), individuals with vascular dementia (VaD) had a higher level of disability and considerably higher rates of cerebrovascular disease, congestive heart failure, hemiplegia, paraplegia, and myocardial infarction, thus increasing both the complexity and costs of management of the disease [3]. Moreover, patients with VaD have been found to have a higher relative risk of death (RR: 2.7, 95% CI, 1.9–3.9) when compared to AD (RR: 1.4, 95% CI, 1.2–1.7) [4]. Interventions that target potentially modifiable risk factors associated with VCI [5], such as minimizing diabetes, hypertension treatment, and avoiding midlife obesity, among others, have been proposed as a way of reducing dementia in low- and middle-income countries [6].

There has been a significant increase of stroke burden in the world especially in developing countries [7]. In addition, the number of people with dementia in Latin American countries is predicted to increase 4-fold in the next 30–35 years [8]. Globally, VaD prevalence estimates range between 0.9 and 3.3% (95% CI, 2.2–4.5) [9]. In developing countries, these estimates vary from 0.7 (95% CI, 0.1–1.3) to 2.1% (95% CI, 1.6–2.7) in those aged over 55 years [10]. In Mexico, the Fogarty Stroke Cohort Study, which recruited relatively young acute post-stroke patients, found a dementia prevalence of 12% three months after stroke [11].

Epidemiological studies report a VaD incidence stratified by age and gender that ranges from 0.99 (95% CI, 0.96–1.02) [12] to 3.4 (95% CI, 2.1–4.9)/1000 person-years [13]. A meta-analysis reported varying rates of post-stroke dementia that ranged from 7.4% in population-based studies of first-ever stroke in individuals with no previous dementia to 41.3% in hospital-based studies of recurrent stroke in which previous dementia diagnosis was included [5]. A greater age-adjusted incidence rate (per 1000 person-years) for VaD has been found in men (12.2) vs women (9.0), along with an increased relative risk with advancing age (RR: 1.6 (95% IC 1.2–2.0) [14].

VCI frequencies and incidence rates have been reported with great variability possibly because of different settings and designs, as well as some studies' neuroimaging accessibility [4, 10, 15]. Specific criteria involving the temporal relationship between a vascular event and the onset of cognitive decline is also to be considered when accounting for differences between studies [15]. Still, experts have been making progress in producing guidelines for a more standardized diagnosis [16, 17], where an umbrella of diagnostic possibilities for VaD is considered. Even if the use of magnetic resonance imaging (MRI) is a gold standard requirement for clinical diagnosis, a definition of possible mild or major VCI (VaD) is appropriate when

neuroimaging is not available and clinically significant cognitive deficits in at least one cognitive domain with or without functional dependence are present [17].

Epidemiological investigation ought to be a first step in the means of attracting attention to dementia subtypes, especially in Latin America, where limited data for stroke and VaD exists [11, 13]. Therefore, the aim of this study is to determine the prevalence and incidence of PVD in older adults using a national representative panel study in Mexico, the Mexican Health and Aging Study (MHAS 2012–2015 waves).

## Materials and methods

### Study population

The study population included participants from the MHAS [18], a national representative panel study of Mexican residents aged ≥50 years with four follow-up waves (2003, 2012, 2015, 2018) of the baseline conducted in 2001. The aim and methodological design of the MHAS is published elsewhere [19]. We analyzed data from the third and fourth MHAS waves collected between October 2012 and December 2015. For ethical approval, the MHAS protocols and instruments were reviewed by the Institutional Review Board of the University of Texas Medical Branch, the National Institute of Statistics and Geography (INEGI for its acronym in Spanish) and the National Institute of Public Health (INSP) in Mexico. MHAS data files and documentation are of public use and available at www.mhasweb.org.

### Sample selection at baseline and follow-up

Fig 1 shows the flowchart of the baseline sample selection from MHAS 2012 wave. A total of n = 14,893 participants aged 50 or older were included. Individuals who answered the interview directly (self-respondents) represented 91.7% (n = 13,651) of the sample, while 8.3% (n = 1,242) were proxy respondents. Based on self-reported history of stroke, we identified n = 338 (2.2%) individuals with stroke and n = 14,552 (97.8%) without stroke. All individuals were further classified as with dementia or cognitively normal based on diagnostic criteria further described.

Fig 2 shows the flowchart of the sample selection at follow-up from the MHAS 2015. Participants were followed an average of 3 years (SD = 0.61). During follow-up n = 837 (5.9%) individuals died ("decedents"), n = 311 (2.2%) refused to answer ("refusals"), and n = 578 (4.1%) could not be contacted ("lost"). The total follow-up sample comprised n = 12,427 (87.8%) individuals, including n = 202 with a history of stroke at baseline, n = 172 new cases of stroke, and n = 12,053 without stroke. Finally, individuals from each group were classified as with dementia or cognitively normal according to the diagnostic criteria further described.

### Cognitive assessment

For self-respondents, the MHAS assesses cognitive function through a modified version [20] of the Cross-Cultural Cognitive Examination (CCCE) [21], which measures performance in eight cognitive domains—verbal learning, delayed memory, attention, constructional praxis, visual memory, verbal fluency, orientation, and processing speed—and has reference norms by age and education. Imputed data were used on cognitive performance for individuals with missing values using a multivariate, regression-based procedure applied by the MHAS team following the same methodology as the Health and Retirement Study [22]. For respondents interviewed by proxy, the MHAS uses a brief version of the Informant Questionnaire on Cognitive Decline in the Elderly (IQCODE) [23]. Proxy interviews are done through an informant, usually a spouse of a close relative when the selected participant is absent or is not healthy

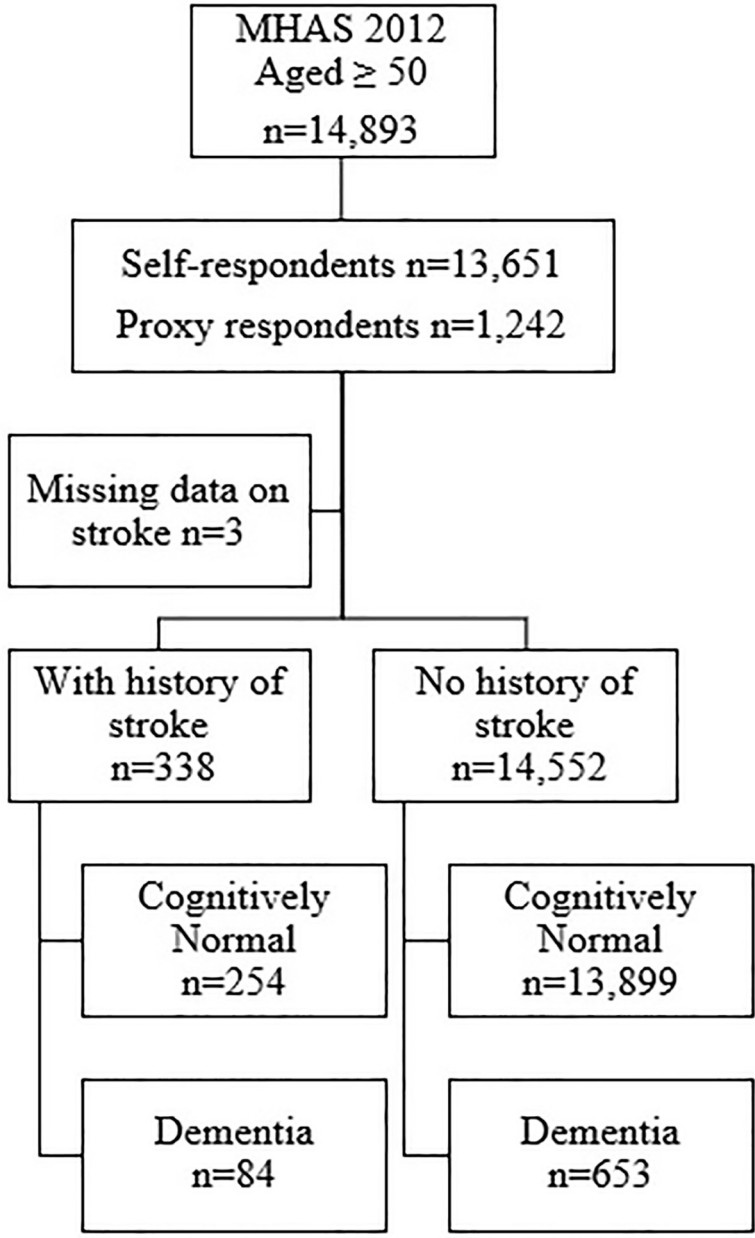

**Fig 1. Flowchart of sample selection at baseline (MHAS 2012).**

enough to complete a direct interview. Self-respondents were classified as having overall dementia if in at least two cognitive domains [24], scores were ≥ 1.5 standard deviations (SD) below the mean based on norms by age and education and also had difficulty performing at least one instrumental activity of daily living (IADL). IADLs included the ability to prepare a meal, go shopping, manage money, or take medications. Pondering the effect of gender on some of the IADLs for the Mexican population (e.g., men do not usually prepare a meal and women do not manage money), individuals that needed help in 1 or more IADL were considered as functionally impaired.

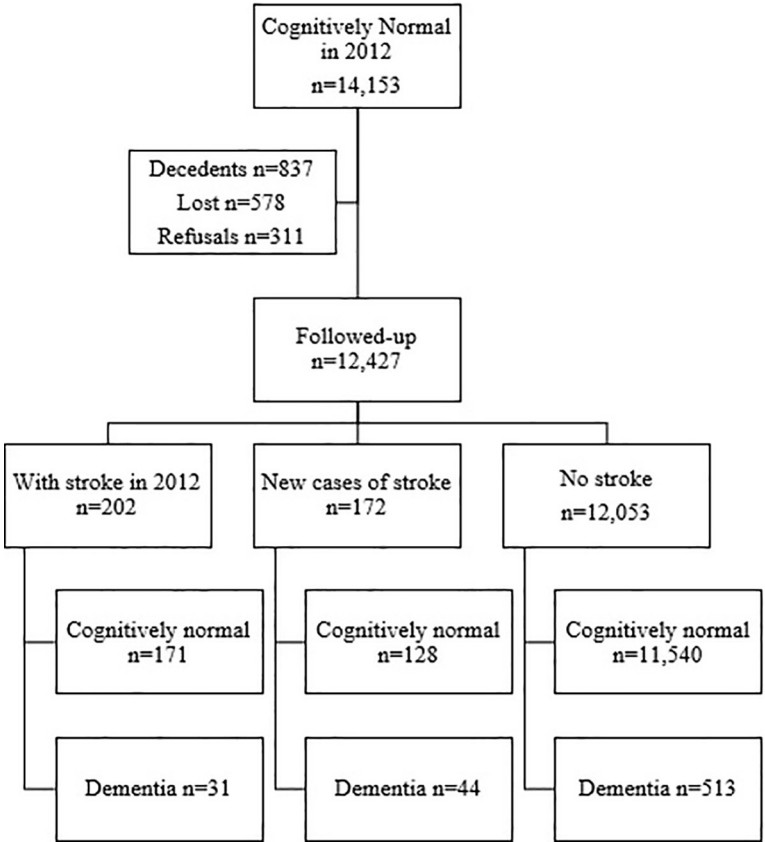

**Fig 2. Flowchart of sample selection at follow-up (MHAS 2015).**

Cognitive performance in those classified as cognitively normal, was no more than one SD below norms in all cognitive domains or ≥1.5 SD below in only one domain, and no IADL limitations were present. For responses collected via proxy-interview, we classified individuals as having overall dementia if IQCODE scores were ≥ 3.4 and < 3.4 for cognitively normal individuals, as recommended [25].

### History of stroke

Participants were classified as having a stroke history based on the question: *Have you ever been diagnosed with stroke by a doctor*? at baseline and follow-up. To correct for possible response bias, we also selected only those individuals who also at least reported one of the following conditions: focal symptoms of stroke (inability to move arms/legs, difficulty speaking/ eating, difficulty with sight/vision, difficulty thinking/expressing him/herself), received rehabilitation therapy or took medications for stroke.

### Diagnostic categories

For this study, possible vascular dementia (PVD) was defined by the combined presence of history of stroke and dementia. To estimate prevalence, n = 84 cases in the 2012 PVD group were considered. The incident PVD group (n = 75) included n = 44 new cases of stroke with dementia and n = 31 cases with stoke history who were cognitively normal at baseline but met criteria for dementia at follow-up. Sociodemographic and health characteristics were not different between these two groups (S1 Table).

The cognitively normal group was defined following the criteria described above based on cognitive domain scores for self-respondents and IQCODE for proxy respondents. For prevalence and incidence analyses we included cognitively normal individuals without stroke history (2012, n = 13,899 and 2015, n = 11,540, respectively).

Cognitively normal cases with stroke and incident dementia cases without stroke were excluded from the analyses.

## Covariates

Sociodemographic characteristics: sex, age as continuous or categorical (50–64, 65–74, $\geq$ 75), years of education as continuous or categorical (0, 1 to 6, $\geq$ 7) following the formative periods in the Mexican education system, and (0 to 3, $\geq$ 4) to classify those in the lowest quartile.

Cardiovascular conditions were self-reported derived from the question: *Has a medical doctor diagnosed you with*: diabetes, hypertension, heart attack or dyslipidemia? We used each disease categorically (yes-no) and as a continuous variable from the sum of all cardiovascular conditions.

Depressive symptoms: the MHAS includes a modified version of the Center for Epidemiological Studies-Depression (CES-D) with nine items (yes/no). We classified respondents with a score $\geq$5 as clinically significant depressive symptoms based on a clinical validation study [26].

Global cognition was constructed as the standardized composite score of the CCCE (sum of scores from each domain) for self-respondents and the IQCODE for proxy respondents.

## Statistical analysis

To examine differences in covariates between diagnostic groups at baseline and follow-up, t-test for continuous variables and Chi-square test for categorical variables, were used. For estimating prevalence and incidence rates the MHAS sampling weights were used [19]. Prevalence rates of PVD were stratified by sex, age, and education. Estimation of total age- and sex-adjusted rates, using data from the Mexican census 2010 as the standard population, was performed [27]. A binomial logistic regression model was used to analyze the association of sociodemographic factors (sex, age, and education), cardiovascular conditions (hypertension, diabetes, and heart attack) on the likelihood of PVD at baseline. For the estimation of incidence rates of PVD, the total time contributed by all participants (person-years) was calculated as follows: responders who remained cognitively normal at follow-up or were present, but refused to participate contributed 3 years (time between MHAS waves 2012 and 2015); those who were lost at follow-up were assigned a contribution of 1.5 years as the mid-point between baseline and follow-up assessment; time contributed by decedents varied depending on the year of death reported by the next of kin at follow-up; time until onset of vascular dementia was defined as the mid-point between baseline and follow-up. The incidence rate of PVD was defined as the number of new (incident) cases during study follow-up divided by the person-time-at risk. To examine baseline predictors of attrition, logistic regression models for each source of attrition: decedents vs. responders, lost vs. responders, and refusals vs. responders, were used (S2 Table). Inverse probability weights (IPW) were employed to account for death and other sources of attrition as competing risks for VaD in the analyses of risk factors [28]. Weights were based on the inverse probability of being observed at follow-up and thus of being alive and uncensored. The rationale behind these weights is that respondents with similar characteristics at baseline to those missing at follow-up, are up-weighted. The probability of being uncensored at follow-up, for each source of attrition (decedents, lost and refusal) versus those alive (completers), was modeled using a logistic regression adjusting for baseline covariates that had shown to influence attrition: sex, age, education, residence (urban-rural), self-

reported diabetes, hypertension, heart attack, and global cognition. Later, the inverse probability weight of survival (1/predicted probability) and the cumulative probability as the product of weights was calculated. Finally, an evaluation of the association between baseline covariates and PVD using IPA-weighted generalized estimating equations (GEE) regression model, was performed. Statistical analyses were using SPSS software for Windows (SPSS Inc., Chicago, IL version 25.0).

## Results

Table 1 provides characteristics of the samples at baseline and follow-up stratified by cognitive status. At baseline, participants with PVD had a mean age of 75.5 (SD 9.5) years, 52.4% were female, and the mean educational level was 4.3 years (SD 4.5). We observed that those with PVD (n = 84) were eleven years older than those in the normal diagnostic group. Persons aged 75 years or older comprise one-half of those with PVD. Sex distribution was not different between groups. On average, PVD individuals had 1.4 years of education less than their normal counterparts, with a significantly lower proportion in the 7+ years of education (17.9%).

**Table 1. Characteristics of the MHAS sample at baseline (2012) and follow-up (2015) by diagnostic group.**

| Characteristics | Baseline (MHAS 2012) | | | Follow-up (MHAS 2015) | | |
|---|---|---|---|---|---|---|
| | Cognitively Normal (n = 13,899) | Possible Vascular Dementia (n = 84) | p-value | Cognitively Normal (n = 11,540) | Possible Vascular Dementia (n = 75) | p-value |
| | Mean (SD) | Mean (SD) | | Mean (SD) | Mean (SD) | |
| | n (%) | n (%) | | n (%) | n (%) | |
| Age, Mean (SD) | 64.8 (9.5) | 75.5 (9.5) | >.001 | 66.8 (9.2) | 75.3 (11.2) | >.001 |
| 50–59 years | 4583 (33.0) | 6 (7.1) | >.001 | 3014 (26.1) | 7 (9.3) | >.001 |
| 60–69 years | 5221 (37.6) | 14 (16.7) | >.001 | 4359 (37.8) | 21 (28.0) | .082 |
| 70–79 years | 2913 (21.0) | 40 (47.6) | >.001 | 2981 (28.8) | 19 (25.3) | .922 |
| 80 + years | 1182 (8.5) | 24 (28.6) | >.001 | 1186 (10.3) | 28 (37.3) | >.001 |
| Sex (female) | 7831 (56.3) | 44 (52.4) | .466 | 6551 (56.8) | 43 (57.3) | .917 |
| Education, years (SD) | 5.7 (4.7) | 4.3 (4.5) | >.001 | 5.7 (4.6) | 3.2 (4.4) | >.001 |
| No education | 2456 (17.7) | 20 (23.8) | .142 | 1959 (17.0) | 26 (34.7) | >.001 |
| 1 to 6 years | 7167 (51.6) | 49 (58.3) | .216 | 6004 (52.0) | 40 (53.3) | .822 |
| 7 + years | 4276 (30.8) | 15 (17.9) | .011 | 3577 (31.0) | 9 (12.0) | >.001 |
| Hypertension | 5928 (42.7) | 54 (64.3) | >.001 | 5449 (47.3) | 53 (70.7) | >.001 |
| Diabetes | 3107 (22.4) | 25 (29.8) | .105 | 2839 (24.6) | 25 (33.3) | .082 |
| Heart attack | 446 (3.2) | 17 (20.2) | >.001 | 397 (3.4) | 13 (17.3) | .>001 |
| Vascular conditions* | 0.7 (0.8) | 1.1 (0.9) | >.001 | 0.8 (0.8) | 1.2 (0.9) | >.001 |
| Depressive symptoms** | 4066 (31.1) | 18 (64.3) | >.001 | 3272 (29.4) | 29 (70.7) | >.001 |
| Global cognition | 0.1 (0.9) | -1.3 (0.8) | >.001 | 0.1 (1.0) | -1.2 (0.8) | >.001 |
| Self-respondents | 13068 (94) | 28 (33.3) | >.001 | 11122 (96.4) | 41 (54.7) | >.001 |
| Proxy respondents | 831 (6.0) | 56 (66.7) | >.001 | 418 (3.6) | 34 (45.3) | >.001 |

P-value from t-test for continuous variables and Chi-square for categorical variables. Values in parentheses are weighted percentages derived using the MHAS sampling weights.

MHAS = Mexican Health and Aging Study.

*Vascular conditions = sum of hypertension, diabetes, and heart attack.

**Significant depressive symptoms ($\geq$5) are presented for self-respondents only (2012 n = 13096; 2015 n = 11162), proxy respondents did not complete the depressive symptoms scale based on the study questionnaire.

**Table 2. Prevalence estimates of possible vascular dementia by sex, age, and education from the MHAS (2012).**

| | Total | | Males | | Females | |
|---|---|---|---|---|---|---|
| | Prevalence estimate | 95% CI | Prevalence estimate | 95% CI | Prevalence estimate | 95% CI |
| Age, y | | | | | | |
| 50–59 | 0.1 | (0.1–0.3[a]) | 0.3 | (0.2–0.7) | 0.0 | - |
| 60–69 | 0.3 | (0.2–0.5) | 0.2 | (0.1–0.5) | 0.3 | (0.1–0.6) |
| 70–79 | 1.4 | (1.0–1.8) | 1.4 | (0.9–2.2) | 1.3 | (0.9–2.0) |
| 80+ | 2.0 | (1.3–2.9) | 1.6 | (0.8–3.0) | 2.3 | (0.1–4.0) |
| Education, y | | | | | | |
| 0 | 0.8 | (0.5–1.2) | 0.9 | (0.5–1.2) | 0.7 | (0.4–1.3) |
| 1–6 | 0.7 | (0.5–0.9) | 0.7 | (0.4–1.0) | 0.7 | (0.5–1.0) |
| 7+ | 0.3 | (0.2–0.6) | 0.5 | (0.3–0.9) | 0.2 | (0.1–0.5) |
| Total crude | 0.6 | (0.5–0.8) | 0.7 | (0.5–0.9) | 0.6 | (0.4–0.8) |
| Total standardized* | 0.5 | (0.5–0.5[b]) | 0.5 | (0.5–0.6) | 0.5 | (0.5–0.5[c]) |

MHAS = Mexican Health and Aging Study. CI = confidence interval. Values are weighted percentages (95% CI) derived using the MHAS sampling weights.

*Standardized = age- and sex-standardized to the Mexican population (2010), a: (0.06–0.28), b: (0.50–0.51), c: (0.48–0.50).

In addition, hypertension (64.3%) and heart attack (20.2%) were the most frequent cardiovascular conditions among PVD individuals ($p > .001$). Still in the PVD group, diabetes rates were higher (29.8% vs. 22.4%), but not statistically significant ($p = 0.105$). Excluding stroke, individuals with PVD had on average 1.1 (SD 0.9) vascular conditions compared to 0.7 (SD 0.8) in the cognitively normal group ($p < .001$). For the analysis of depressive symptoms only self-respondents (n = 28) were included given that proxy participants did not complete the depression scale. The proportion of individuals with significant depressive symptoms in the PVD group (64.3%) was twice as high as their normal counterparts (31.1%). Global cognition scores were significantly lower (-1.3, SD 0.8) in the PVD group compared to normal individuals (0.1, SD 0.9) ($p < .001$). Regarding the type of interview through which participants completed the MHAS survey, two-thirds of the PVD sample were proxy respondents, compared to 6% of the normal sample ($p < .001$).

At follow-up (Table 1), participants with incident PVD (n = 75) showed similar characteristics as the prevalent baseline PVD group. Compared to those cognitively normal, incident PVD adults were on average 8.5 years older, more than half of the group was in the oldest group (≥75years); they were 2.5 years less educated with no education in nearly 35% of the group. Rates of hypertension (70.7% vs.47.3%) and heart attack (17.3% vs. 3.4%) were significantly higher (p < .001). Diabetes was also higher (33.3% vs. 24.6%) in the incident PVD group but the difference did not reach statistical significance (p = 0.082). On average, the incident PVD group had 1.2 (SD 0.9) cardiovascular conditions (excluding stroke) while those cognitively normal had 0.8 (SD 0.8) ($p < .001$). Considering self-respondents only (n = 67), the proportion of individuals with significant depressive symptoms was significantly higher among incident PVD individuals (70.7%) compared to the normal group (29.4%) ($p < .001$). On average, global cognition scores were significantly lower (-1.2 SD 0.8) in the incident PVD group compared to normal individuals (0.1 SD 1.0) ($p < .001$). Finally, the proportion of the incident PVD sample in MHAS represented by a proxy respondent was 45.3% vs. 3.6% compared to the cognitively normal sample ($p < 0.001$)

Table 2 provides prevalence rates of PVD for men and women by age, sex, and education. The total crude prevalence of PVD was 0.6% (95% CI, 0.5–0.8). Prevalence increased with age reaching 2.0% (95% CI, 1.3–2.9) among individuals aged 80 and older, and decreased with

higher educational level from 0.8 (95% CI, 0.5–1.2) among those with no schooling, to 0.3% (95% CI, 0.2–0.6) for those with 7 or more years of education. Sex differences were only significant among the youngest group (50 to 59 years) with 6 cases for men and no cases for women. We found an overall age- and sex-adjusted PVD prevalence of 0.5 with no variations between males and females.

Table 3 shows rates of incident PVD by sex and age. A total of 87 new cases comprises 75 observed individuals and 12 estimated cases from the different sources of attrition (decedents = 8, lost = 3, and refuse = 1). Eighty-seven new cases represented an overall 2.2 (95% CI, 1.7–2.6) incidence rate of PVD per 1,000 person-years, the observed 75 cases represented an incidence rate of 1.9 (95% CI, 1.5–2.5). Incidence increased progressively with age, reaching an overall rate of 9.4 (95% CI, 6.3–13.6) per 1,000 person-years for individuals aged >80 years. Total crude incidence rates of PVD were not different between males 2.2 (95% CI, 1.5–2.9) and females 2.2 (95% CI 1.6–2.8). Although, increasing rates of incident vascular dementia with age were slightly higher for females than males between ages 60 to 79 years, at age 80, males showed a significant increase with a rate of 12.3 (95% CI, 7.2–19.7) per 1,000 person-years compared to females who had a rate of 7.0 (95% CI, 3.5–12.4). We found an overall standardized age- and sex-adjusted PVD incidence of 2.0 with no variations between males and females.

Table 4 shows the results of the full logistic models predicting incident PVD using GEE with IPA-weights to account for attrition. The first model includes all individuals who participated in the survey through direct and proxy interviews (n = 11,615). We estimated a second model for self-respondents only (n = 11,039) to analyze the association of depressive symptoms in 67 incident cases of PVD. Results showed that being male tended to be associated with a higher risk of PVD, but estimates were only marginally significant ($p$ = 0.081) among self-respondents (OR 1.6, 95% CI, 0.9–2.7). At age 75 years and older individuals had 3.6 greater odds of incident PVD compared to the youngest group (50–64 years). Being in the lowest quartile of the education distribution (0–3 years) increased odds of incident PVD (OR 2.8, 95% CI, 1.5–5.2). The presence of hypertension (OR 2.6, 95% CI, 1.6–4.4), lower global cognition (OR 0.7, 95% CI, 0.5–1.0), and depressive symptoms (OR 2.6, 95% CI, 1.5–4.3) at baseline were associated with increased risk of incident PVD.

**Table 3. Incidence rates of possible vascular dementia from the MHAS 2015.**

| Age | Total | | | | Males | | | | Females | | | |
|---|---|---|---|---|---|---|---|---|---|---|---|---|
| | Person-years at risk | Cases | Incidence rate/1000 | 95% CI | Person-years at risk | Cases | Incidence rate/1000 | 95% CI | Person-years at risk | Cases | Incidence rate/1000 | 95% CI |
| 50–59 | 13714 | 9 | 0.6 | (0.3–1.2) | 5253 | 5 | 0.8 | (0.0–1.9) | 8461 | 4 | 0.5 | (0.1–1.2) |
| 60–69 | 15327 | 27 | 1.8 | (1.2–2.5) | 7097 | 9 | 1.3 | (0.1–2.4) | 8230 | 18 | 2.3 | (1.4–3.6) |
| 70–79 | 8260 | 23 | 2.7 | (1.8–4.2) | 3826 | 8 | 2.1 | (0.1–4.1) | 4433 | 15 | 3.2 | (1.7–5.3) |
| 80+ | 2964 | 28 | 9.4 | (6.3–13.6) | 1383 | 17 | 12.3 | (7.2–19.7) | 1581 | 11 | 7.0 | (3.5–12.4) |
| Total crude | 40264 | 87 | 2.2 | (1.7–2.6) | 17559 | 38 | 2.2 | (1.5–2.9) | 22705 | 49 | 2.2 | (1.6–2.8) |
| Total standardized* | - | - | 2.0 | (1.3–2.7) | - | - | 2.0 | (0.1–3.0) | - | - | 2.0 | (1.0–2.9) |

CI = confidence intervals; MHAS = Mexican Health and Aging Study. Values are weighted percentages (95% CI) derived using the MHAS sampling weights.

*Standardized = age- and sex-standardized to the Mexican population (2010).

**Table 4. Generalized estimation equations regression models for predictors of possible vascular dementia using IPA-weights.**

| | Incident Possible Vascular Dementia*[1] Self and Proxy Respondents | | | Incident Possible Vascular Dementia*[2] Self-Respondents | | |
|---|---|---|---|---|---|---|
| **Baseline predictors** | **OR** | **p-value** | **95% CI** | **OR** | **p-value** | **95% CI** |
| Sex (male) | 1.3 | .278 | (0.8–2.1) | 1.6 | .081 | (0.9–2.7) |
| Age | | | | | | |
| • 50–64 (ref) | 1 | | | 1 | | |
| • 65–74 | 1.1 | .792 | (0.6–2.1) | 1.0 | .950 | (0.5–3.7) |
| • 75 + | 3.6 | >.001 | (1.9–6.9) | 2.7 | .005 | (1.3–5.3) |
| Education | | | | | | |
| • 0 to 3 | 2.8 | >.001 | (1.5–5.2) | 2.0 | .025 | (1.1–3.7) |
| • 4 or more (ref) | 1 | | | 1 | | |
| Diabetes | 1.6 | .084 | (0.9–2.5) | 1.4 | .277 | (0.8–2.3) |
| Heart Attack | 1.1 | .936 | (0.3–3.6) | 0.7 | .566 | (0.2–2.9) |
| Hypertension | 2.6 | >.001 | (1.6–4.4) | 2.3 | .003 | (1.3–4.0) |
| Global cognition | 0.7 | .036 | (0.5–1.0) | 0.6 | .003 | (0.4–0.8) |
| Depressive Symptoms | NA | NA | | 2.6 | >.001 | (1.5–4.3) |
| Observations | 11615 | | | 11039 | | |

*[1] Estimates considering all cases of incident possible vascular dementia (PVD, n = 75)

*[2] Estimates considering cases of incident PVD with information on depressive symptoms (PVD. n = 67) and excluding proxy respondents who did not complete the depressive symptoms scale based on the proxy study questionnaire. IPA = inverse-probability-of-attrition; OR = odd ratio; CI = confidence interval; MHAS = Mexican Health and Aging Study; OR = odd ratio.

## Discussion

The prevalence of PVD in Mexican adults aged 50 years or older in the 2012-MHAS wave was 0.6% (95% CI, 0.5–0.8). A systematic analysis performed by Kalaria et. al [10], reporting VaD prevalence in developing countries, also found low VaD estimates; 0.7% (95 CI, 0.1–1.3) in an analysis of four studies in Taiwan where there is no mention of neuroimaging data and 1.1% (95% CI, 0.2–1.9) in a sub analysis of five rural and urban studies in India, where only two studies [29, 30] made use of brain imaging. In this same systematic analysis, authors reviewed data from 12 centers and concluded that only 26% of cases of dementia fulfilled the National Institute of Neurological Disorders and Stroke and Association Internationale pour la Recherché et l'Enseignement en Neurosciences (NINDS-AIREN) criteria for VaD [31]. Thus, a higher 2.1% (95% CI; 1.6–2.7) prevalence was found in Venezuela [32] and even a 6% frequency was reported in another study in Israel [33]. The latter used the Diagnostic and Statistical Manual of Mental Disorders Fourth Edition (DSM-IV) criteria [34] for dementia diagnosis and made VaD diagnosis at hospital discharge after stroke, while the Maracaibo study used the NINDS-AIREN criteria [31] for VaD, included several more neuropsychological cognitive tests, and reported 67% access to neuroimaging data [32].

Age is the main risk factor for any kind of dementia, including cognitive impairment of vascular origin. Based on results from the Maracaibo study in Latin America, Molero et al. [32] also found an increasing pattern of VaD with age, with a frequency of 0.2% (95% CI, 0.1–0.7), 2.0% (95% CI, 1.8–3.3), and 5.2% (95% CI, 3.4–7.8) in adults aged 55–64, 65–74, and 75–84, respectively, which is comparable to our findings. We also found no significant sex variations for PVD prevalence, which is similar to what other Latin-American studies found [11, 32].

In the present study PVD prevalence decreased from 0.8 (95% CI, 0.5–1.2) among individuals with no education, to 0.3% (95% CI, 0.2–0.6) for those with 7 or more years of education. A systematic review reported that the presence of lower educational level increased the risk of VaD 2.5 times (OR 2.5, 95% CI, 1.8–3.4, p < .001) [5]. Similarly, and after adjusting for confounding cardiovascular variables, the Rotterdam Study [35] found that only the least educated (primary education) were at risk for VaD (RR: 2.1, 95% CI, 1.0–4.5). A lack of cognitive reserve, where pre-existing mechanisms allow for neural compensation and favor resilience when coping with the damage caused by vascular pathology, can be a possible explanation in situations where a lower educational level is associated with greater dementia risk [36, 37].

Our rate for PVD incidence [2.2 (95% CI, 1.7–2.6)] was lower than that reported in the Maracaibo study [13], in which a 3.4 (95% CI, 2.2–4.9) per 1000 person-years rate was found. Access to neuroimaging was substantial in this last study, which could again explain the underestimation of VaD cases found in our study, particularly involving subcortical disease, small lacunar brain infarctions, and mixed dementia cases, which are considered as part of the umbrella of diagnostic possibilities for VaD [17]. Another study in the UK [12], which described a lower VaD incidence rate compared to our findings [0.99/1,000 person-years (95% CI, 0.96–1.02)], used an algorithm to identify individuals' first-time VaD diagnosis based on DSM-IV [34], NINCDS-ADRDA [38] or NINDS-AIREN [31] dementia criteria. Only 15% of participants in this last study complied with diagnosis after neuroimaging. Methodological differences between studies, such as the use of different diagnostic criteria, the inclusion of additional cognitive evaluation tests, and neuroimaging availability, could account for variability in results.

An increasing VaD incidence with age of 0.8 (95% CI, 0.2–2.3), 3.8 (95% CI, 1.9–6.9), and 8.9 (95% CI, 4.1–16.9)/1,000 person-years, in adults 55–64, 64–74, and 75–84 years of age, respectively, was described by Maestre et.al [13]. These findings are comparable to our study, given that incidence rates reached an overall 9.4 per 1,000 person-years in individuals aged >80 years. We also observed that males aged 80 years or older, showed greater incidence rates when compared to females [12.3 (95% CI, 7.2–19.7) vs 7.0 (95% CI, 3.5–12.4) per 1,000 person-years], which is also consistent with what other studies have shown [12, 14, 39].

The presence of depressive symptoms (OR = 2.6, 95% CI, 1.5–4.3) and hypertension (OR = 2.6, 95% CI, 1.6–4.4) at baseline were associated with an increased risk of incident PVD. It has been proposed that late life-depression is a risk factor for VaD as it is for AD [1]. A meta-analysis [40] reported a 2.5 (95% CI, 1.8–3.6) and 1.7 (95% CI, 1.4–1.9) greater risk for VaD and AD, respectively, associated with late-life depression. Moreover, poorer health and a higher burden of cardiovascular and cerebrovascular disease have also been associated to depression in older age [41, 42]. Barnes, et.al., in a retrospective cohort study found that chronic depression during the life course may be associated with an increased risk of dementia, particularly VaD [43]. Vascular abnormalities, specifically white matter changes and small vessel disease possibly playing a contributing role for dementia development, have been observed on brain imaging of depressed patients [44]. Other studies have found similar results concerning high blood pressure and VaD, with an OR of 2.0 (95% CI, 1.3–2.9) in the Hisayama study [14] and 1.2 (95% CI, 1.1–1.3) in a report by Imfeld et.al [12]. Hypertension is a major risk factor for stroke, thus linking it to VCI [7]. Recent reports have found that high blood pressure throughout midlife, increases the risk of dementia alone, even without stroke [6].

We acknowledge several limitations. First, information was obtained through a survey, with results that can only be applied to the Mexican population. Due to the nature of the survey's IADLs evaluation, a clear differentiation between disability due to stroke complications or because of cognitive impairment alone, is not allowed. Second, as in most epidemiological surveys, neuroimaging studies were not systematically performed in the MHAS. The exclusion of

new dementia cases without a history of stroke but with small vessel disease or silent lacunar strokes that can be identified through neuroimaging, could have also played an important role in the underestimation of VaD cases. Third, the cognitive evaluation test used in the MHAS, although proper in a cross-cultural context, could have also contributed to an underestimation of cases, since its use does not allow a clear identification of subcortical cognitive domains. Lastly, two-thirds of the sample included in the PVD group were diagnosed using answers of proxy respondents, which may also have made PVD criterion less sensitive for detecting new cases.

One of the strengths of our study is that it is the first study that shows PVD incidence rates in a large sample of Mexican residents. It is a longitudinal, representative, study that includes adults 50 years or older and shows an overview of important sociodemographic risk factors that mirror the health situation faced in developing countries, where educational policies, health behaviors, and health care practices might be different from that in high-income countries. Additionally, common cardiovascular risk factors are present predominantly in Latin America, highlighting the need for prevention strategies [45].

## Conclusions

These data provide new estimates of PVD prevalence and incidence in the Mexican population. Males aged 80 years or older showed a greater incidence rate when compared to females, which is comparable to the mean of previous incidence estimates from other studies. Vascular dementia prevention strategies should focus on potentially modifiable risk factors.

## Supporting information

**S1 Table. General characteristics of incident possible vascular dementia by time of stroke register.** S1 Table provides general characteristics of individuals with incident possible vascular dementia comparing those who reported history of stroke in 2012 and 2015. Results show no significant differences in sociodemographic, cardiovascular conditions, depressive symptoms (≥5), and global cognition among groups.
(PDF)

**S2 Table. Logistic regression models predicting attrition at follow-up.** S2 Table shows the results of the full logistic regression models for each source of attrition compared to responders. Being male, older age, having history of diabetes and heart attack and lower global cognition were predictors of mortality. Attrition due to lost was associated with lower age, higher education, living in urban areas, having history of stroke, lower hypertension, and lower global cognition. Refusals were predicted by lower age, higher education, and lower rates of hypertension.
(PDF)

## Acknowledgments

All authors had substantial contributions in the conception, design, analysis, or interpretation of the work. All authors contributed to the draft and revision of the content, and final approval version to be published.

## Author Contributions

**Conceptualization:** Alberto J. Mimenza-Alvarado, José A. Avila-Funes, Sara G. Aguilar-Navarro.

**Data curation:** Sara G. Yeverino-Castro, Silvia Mejía-Arango, Alberto J. Mimenza-Alvarado, Sara G. Aguilar-Navarro.

**Formal analysis:** Sara G. Yeverino-Castro, Silvia Mejía-Arango, Carlos Cantú-Brito, Sara G. Aguilar-Navarro.

**Funding acquisition:** Sara G. Aguilar-Navarro.

**Investigation:** Silvia Mejía-Arango, Alberto J. Mimenza-Alvarado, Carlos Cantú-Brito, Sara G. Aguilar-Navarro.

**Methodology:** Silvia Mejía-Arango, Alberto J. Mimenza-Alvarado, Carlos Cantú-Brito, Sara G. Aguilar-Navarro.

**Project administration:** Sara G. Aguilar-Navarro.

**Resources:** Sara G. Yeverino-Castro.

**Software:** Sara G. Yeverino-Castro, Silvia Mejía-Arango.

**Supervision:** Silvia Mejía-Arango, Carlos Cantú-Brito, José A. Avila-Funes.

**Validation:** Silvia Mejía-Arango, Carlos Cantú-Brito, José A. Avila-Funes, Sara G. Aguilar-Navarro.

**Writing – original draft:** Sara G. Yeverino-Castro, Silvia Mejía-Arango, José A. Avila-Funes.

**Writing – review & editing:** Alberto J. Mimenza-Alvarado, Carlos Cantú-Brito, José A. Avila-Funes, Sara G. Aguilar-Navarro.

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
