## [Decision Letter · Decision Letter 0]

3 Mar 2021

PONE-D-20-39225

Prevalence and incidence of possible vascular dementia among older Mexican adults: analysis of the Mexican Health and Aging Study

PLOS ONE

Dear Dr. Aguilar-Navarro,

Thank you for submitting your manuscript to PLOS ONE. After careful consideration, we feel that it has merit but does not fully meet PLOS ONE’s publication criteria as it currently stands. Therefore, we invite you to submit a revised version of the manuscript that addresses the points raised during the review process.

We look forward to receiving your revised manuscript.

Kind regards,

Claudia K. Suemoto

Academic Editor

PLOS ONE

Journal Requirements:

Reviewers' comments:

Reviewer's Responses to Questions

**Comments to the Author**

1. Is the manuscript technically sound, and do the data support the conclusions?

Reviewer #1: Yes

Reviewer #2: Yes

2. Has the statistical analysis been performed appropriately and rigorously? 

Reviewer #1: Yes

Reviewer #2: No

3. Have the authors made all data underlying the findings in their manuscript fully available?

Reviewer #1: Yes

Reviewer #2: Yes

4. Is the manuscript presented in an intelligible fashion and written in standard English?

Reviewer #1: No

Reviewer #2: Yes

5. Review Comments to the Author

Reviewer #1: This manuscript subject stands out a great health concern since dementia population will increase in the next years in Latin America. The topic is interesting and here I present some suggestions for the authors to make the manuscript more suitable and complete for publication:

1. Introduction

Cardiovascular diseases, including stroke, are the leading cause of death globally. Its consequences bring a significant burden to its carriers and we can identify post-stroke cognitive impairment as being one of the greatest onuses, as it affects labor independence and quality of life. As dementia is the 5th cause of mortality in the world, the authors should bring information about dementia and vascular dementia mortality rates and burden to the health system (costs, for example). This might increase the relevance of recognizing the prevalence and incidence of vascular cognitive impairment, so interventions are targeted to potentially modifiable risk factors associated with VCI.

Still in the introduction, pg 11, line 81, I would quote that “a meta-analysis reported rates of post-stroke dementia that ranged from 7.4% […] to 41.3% […]”.

2. Methods

The methods and study design are appropriate for the questions and claims of this study. When reading the section “Cognitive assessment”, I suggest the authors to cite the reference for the criterion of overall dementia for self-respondents (pg 7, line 134).

Regarding IADLs impairment, some cultural influences may affect the interpretation of this matter. One to consider is the effect of gender on IADLs for the Latin population (some men do not usually prepare a meal and some women may not manage money) – was that considered? Still in IADLs impairment, it can happen due to stroke complications (focal symptoms), as difficult to move arms/legs (so the patient will have difficult in cooking, for example). Was it possible to separate IADL disability due to cognitive impairment and not due to stroke complications? If these two points could not be considered, the authors should cite them as limitations.

Still in the method section, as this study involved human participants, it is important to include ethical approval, as the name of the institutional review board or ethics committee that approved the research. The MHAS protocols and instruments were approved by the Institutional Review Board of the University of Texas Medical Branch, the National Institute of Statistics and Geography (INEGI for its acronym in Spanish) and the National Institute of Public Health (INSP) in Mexico (In pg 5, line 101) – is this the ethical approval? This information needs to be more specific.

3. Figures, tables and results

The figures and tables are clearly presented and correctly labeled. I suggest that, throughout the text, the results can be cited with only one decimal place, as an example on pg 22 line 294: “At age 75 years and older individuals had 3.6 greater odds of incident PVD…” and line 295: “Being in the lowest quartile of the education distribution increased in 2.8 the odds of incident PVD”.

4. Conclusions

The overall rate of PVD incidence found was lower of that reported in other studies and authors pointed out that methodological differences between studies could account for variability in results. The authors should mention which differences there are that could explain this variability, with references (e.g. use of different PVD criteria and cognition scales?).

In the limitation section of the discussion, the fact two thirds of the PVD population were diagnosed by proxy respondents may have made the criterion for detecting PVD less sensitive, so the authors should mention that. Second, the exclusion of new cases of dementia without history of stroke is also a limitation and must be explained through the manuscript, such as patients with cerebral small vessel disease without typical stroke history and evolving with dementia.

I suggest the authors make a careful analysis of grammatical and spelling errors, as: Pg 17, line 316 – “found an increasing pattern”, Pg 18, line 332: “hypothesis where pre-existing mechanisms” and Pg 18, line 340: methodological.

Reviewer #2: The study by Yeverino-Castro et al. reported the prevalence and incidence of vascular dementia in the MHAS.

The manuscript is well-written and the findings are consistent with other studies performed previously.

I have only 2 major comments:

1. was the weights of MHAS that are needed to have prevalence and incidence rates representative of the Mexican population used?

2. In the Discussion, the authors need to discuss how using a criteria attached to the presence of stroke influence the prevalence and incidence of vascular dementia. They need to make clearer that the findings probably understimante the rates of VaD in this study.

6. PLOS authors have the option to publish the peer review history of their article (what does this mean?). If published, this will include your full peer review and any attached files.

Reviewer #1: No

Reviewer #2: No

---

## [Author Response · Author response to Decision Letter 0]

14 May 2021

April 8, 2021

Claudia K. Suemoto

Academic Editor

PLOS ONE Journal 

We would like to thank the editor for his helpful suggestions. We also thank the reviewers for their generous comments on the manuscript. We have addressed all the suggestions in our current version which we allow ourselves to review and edit with the direction of your comments.

In this letter, our responses are written in purple. We have included a marked-up copy (pdf) of the manuscript where changes are highlighted in yellow.

In response to Journal’s additional requirements:

1. Please review your reference list to ensure that it is complete and correct. 

- Thank you for identifying this. We have corrected 2 references [22 and 28] that had a broken e-link. 

- Thank you for identifying the problem. We have corrected the Title page to accommodate PLOS ONE, Title, Autor, Affiliations formatting guidelines. We included a full and short title (page 1, lines 1-3) and used symbols to indicate author contributions and included the corresponding author’s e-mail address (page 1, line 14 and 15).

- We added an author, José A. Ávila Funes (page 1, line 5), who was not included in the initial submission. We acknowledge our error but want to correct it, as this author meets criteria for authorship, and must be included. 

- We removed the key word section in the abstract (originally on page 3, line 51 and 52).

- We also corrected figure citations according to guidelines (page 6, line 114 and 121).

3. We note that you have included the phrase “data not shown” in your manuscript. 

- We have removed this sentence originally located on pg. 11, lines 228-230 of the unrevised manuscript. The data was not a core part of the research being presented in our study.

4. In your Data Availability statement, you have not specified where the minimal data set underlying the results described in your manuscript can be found. 

- We have added, in the manuscript’s methodology section (page 6, lines 109 and 110) as well as in the revised cover letter, an e-link to access the MHAS data and documentation, which are of public use and available at www.mhasweb.org. If specific data is required, please let us know, so we can share the requested information.

In response to Reviewer #1 comments:

5. Introduction: (…the authors should bring information about dementia and vascular dementia mortality rates and burden to the health system (costs, for example).

- Thank you for your recommendation. To increase the relevance of recognizing vascular cognitive impairment we have updated the first paragraph with information on dementia’s mortality and cost of care (page 4, lines 61-66).

6. Still in the introduction, pg 11, line 81, I would quote that “a meta-analysis reported rates of post-stroke dementia that ranged from 7.4% […] to 41.3% […]”.

- We have made this correction, page 4, line 79.

7. Methods: When reading the section “Cognitive assessment”, I suggest the authors to cite the reference for the criterion of overall dementia for self-respondents (pg 7, line 134).

- We appreciate that you have found this. We have added the missing reference (page 7, line 142).

8. Methods: The effect of gender on IADLs for the Latin population (some men do not usually prepare a meal and some women may not manage money) – was that considered? 

- We are very thankful that you have noticed this. We have added a comment about the effect on gender in the Mexican population on page 7, lines 145-147). 

9. Methods: Was it possible to separate IADL disability due to cognitive impairment and not due to stroke complications?

- We appreciate your comment. We have included this issue on the limitations section of the study (page 20, lines 380-382).

10. Methods: (In pg 5, line 101) – is this the ethical approval? This information needs to be more specific.

- As correctly noted in page 6 line 106, the Institutions mentioned are responsible for ethical approval of the MHAS protocol.

11. Figures, tables, and results. I suggest that, throughout the text, the results can be cited with only one decimal place. “At age 75 years and older individuals had 3.6 greater odds of incident PVD…” and line 295: “Being in the lowest quartile of the education distribution increased in 2.8 the odds of incident PVD”.

- We are thankful for your suggestion. Decimals on page 16, line 296 and 297, have been modified, as well as all decimals in figures, tables, and results throughout the manuscript.

- For decimals with minor numeric intervals (e.g., 0.48 – 0.50) we chose to include the exact 2 decimal digit at the corresponding table’s footnote (page 14, 272, 273) and at one citation (page 4, line 78).

12. Conclusions. The authors should mention which differences there are that could explain this variability, with references (e.g., use of different PVD criteria and cognition scales?).

- We have made this distinction by explaining, in the Introduction (page 5, lines 84-93) section of the Manuscript, how methodological differences between studies, such as the use of different diagnostic criteria, the inclusion of additional cognitive evaluation tests, and studies’ neuroimaging availability, could account for variability in results.

13. Conclusions. In the limitation section of the discussion, the fact two thirds of the PVD population were diagnosed by proxy respondents may have made the criterion for detecting PVD less sensitive, so the authors should mention that.

- We have included a statement regarding this issue on page 20, lines 388-390.

14. Conclusions. Second, the exclusion of new cases of dementia without history of stroke is also a limitation and must be explained through the manuscript, such as patients with cerebral small vessel disease without typical stroke history and evolving with dementia.

- We have learned a lot from your comments and are deeply grateful for your them. We have added a statement about this issue in the limitations section (page 20, lines 383-386) of our study. 

15. I suggest the authors make a careful analysis of grammatical and spelling errors, as: Pg 17, line 316 – “found an increasing pattern”, Pg 18, line 332: “hypothesis where pre-existing mechanisms” and Pg 18, line 340: methodological.

- We are very thankful for noticing this. A careful grammatical and spelling analysis has been made and errors have been corrected (page 18, line 330 and 340).

In response to Reviewer #2 comments:

16. Were the weights of MHAS that are needed to have prevalence and incidence rates representative of the Mexican population used?

- The 2001 baseline survey of the MHAS includes a nationally representative sample of individuals born prior to 1951, that is the population aged 50 or older as of 2001. The sample for the MHAS baseline was selected from residents of both rural and urban areas and distributed in all 32 states of the country in urban and rural areas. A new sample of adults born between 1952-1962 was added in 2012. This information is explained elsewhere, and we added the reference for this on page 6 line 104 and 105.

17. In the Discussion, the authors need to discuss how using a criterion attached to the presence of stroke influence the prevalence and incidence of vascular dementia. They need to make clearer that the findings probably underestimate the rates of VaD in this study.

- We agree and are thankful for your comments. We believe we have addressed this issue in the Discussion section (page 19, lines 346-349) of our study, as well as in page 20 lines 382-390.

We believe the manuscript is now suitable por publication in PLOS ONE.

Sincerely. 

Corresponding Author

---

## [Decision Letter · Decision Letter 1]

7 Jun 2021

PONE-D-20-39225R1

Prevalence and incidence of possible vascular dementia among Mexican older adults: analysis of the Mexican Health and Aging Study

PLOS ONE

Dear Dr. Aguilar-Navarro,

Thank you for submitting your manuscript to PLOS ONE. After careful consideration, we feel that it has merit but does not fully meet PLOS ONE’s publication criteria as it currently stands. Therefore, we invite you to submit a revised version of the manuscript that addresses the points raised during the review process.

We look forward to receiving your revised manuscript.

Kind regards,

Claudia K. Suemoto

Academic Editor

PLOS ONE

Journal Requirements:

Reviewers' comments:

Reviewer's Responses to Questions

**Comments to the Author**

1. If the authors have adequately addressed your comments raised in a previous round of review and you feel that this manuscript is now acceptable for publication, you may indicate that here to bypass the “Comments to the Author” section, enter your conflict of interest statement in the “Confidential to Editor” section, and submit your "Accept" recommendation.

Reviewer #1: All comments have been addressed

Reviewer #2: (No Response)

2. Is the manuscript technically sound, and do the data support the conclusions?

Reviewer #1: Yes

Reviewer #2: Yes

3. Has the statistical analysis been performed appropriately and rigorously? 

Reviewer #1: Yes

Reviewer #2: No

4. Have the authors made all data underlying the findings in their manuscript fully available?

Reviewer #1: Yes

Reviewer #2: Yes

5. Is the manuscript presented in an intelligible fashion and written in standard English?

Reviewer #1: Yes

Reviewer #2: Yes

6. Review Comments to the Author

Reviewer #1: (No Response)

Reviewer #2: In my previous review, I have requested information on the use of sampling weigths.

However, the authors did not answer this comment.

They provided information about the sample procedure.

In epidemioloic sutdies, we cannot evaluate the whole population, so we sample some individuals.

To make the sample representative of the population, we can weight the analyses by the sampling weigths.

This is a common procedure in studies that aim to measure prevalence and incidence rates.

Please, provide information if you used the weights.

If you have not used them, recalculate your analyses.

7. PLOS authors have the option to publish the peer review history of their article (what does this mean?). If published, this will include your full peer review and any attached files.

Reviewer #1: No

Reviewer #2: **Yes: **Claudia Kimie Suemoto

---

## [Author Response · Author response to Decision Letter 1]

8 Jun 2021

Dear: 

Academic Editor

PLOS ONE Journal

We would like to thank academic editor for her helpful suggestions. We also thank the reviewers for their generous comments on the manuscript. We have addressed all the suggestions in our current version which we reviewed and edited accordingly.

In this letter, our responses are written in purple. We have included a marked-up copy of the manuscript where changes are highlighted in yellow.

In response to Journal’s additional requirements:

1. Please review your reference list to ensure that it is complete and correct. 

- Thank you for identifying this. We have corrected a misplaced reference (#17 instead of #15) in page 5, line 94.

- We also noted a misplaced reference on page 6 line 110-111. The web address where information can be downloaded is www.mhas.org. The aim and methodological design of the MHAS is indeed published elsewhere, and reference 19 (https://doi.org/10.1093/ije/dyu263) is correctly placed at page 6 line 106. 

- We added a previously misplaced reference (#45) at page 21 line 399 (previously referenced as #40)

- We also removed any bold formatting we used for references.

In response to reviewer #2: 

1. In my previous review, I have requested information on the use of sampling weights. However, the authors did not answer this comment. They provided information about the sample procedure. In epidemiologic studies, we cannot evaluate the whole population, so we sample some individuals. To make the sample representative of the population, we can weight the analyses by the sampling weights. This is a common procedure in studies that aim to measure prevalence and incidence rates. Please, provide information if you used the weights. If you have not used them, recalculate your analyses.

- We are very thankful of your comment. Yes, we used MHAS sampling weights. We now added a sentence at the statistical analysis section (page 10, line 192-193) including the reference to the paper that explains the MHAS sampling weights. We also added this information to each of the tables where weighted data is presented; Table 1 (page 13, line 244-245), Table 2 (page 15, line 274-275), Table 3 (page 16, lines 292-293), and Supporting information table 1 (S1_table).

We believe the manuscript is now suitable por publication in PLOS ONE.

Sincerely. 

Corresponding Author

Sara G. Aguilar Navarro 

On behalf of all authors.

---

## [Editor Report · Decision Letter 2]

15 Jun 2021

Prevalence and incidence of possible vascular dementia among Mexican older adults: analysis of the Mexican Health and Aging Study

PONE-D-20-39225R2

Dear Dr. Aguilar-Navarro,

We’re pleased to inform you that your manuscript has been judged scientifically suitable for publication and will be formally accepted for publication once it meets all outstanding technical requirements.

Kind regards,

Claudia K. Suemoto

Academic Editor

PLOS ONE
---

## [Editor Report · Acceptance letter]

29 Jun 2021

PONE-D-20-39225R2 

Prevalence and incidence of possible vascular dementia among Mexican older adults: analysis of the Mexican Health and Aging Study 

Dear Dr. Aguilar-Navarro:

I'm pleased to inform you that your manuscript has been deemed suitable for publication in PLOS ONE. Congratulations! Your manuscript is now with our production department. 

Kind regards, 

on behalf of

Dr. Claudia K. Suemoto 

Academic Editor

PLOS ONE